# Analysis of Plasma Dynamics in He-Ne Lasers with Different Gas Ratios

Yuanhao Mao [1,2,†] , Jing Hu [1,2,†], Hongteng Ji [1,2], Shiyu Guan [1,2], Dingbo Chen [1,2], Qiucheng Gong [1,2], Wei Liu [1,2], Xingwu Long [1,2] and Zhongqi Tan [1,2,*]

1   College of Advanced Interdisciplinary Studies, National University of Defense Technology, Changsha 410073, China; maoyh96@outlook.com (Y.M.); 15616218757@163.com (J.H.); tenge-12@foxmail.com (H.J.); guanshiyu0211@outlook.com (S.G.); chendingbo15@nudt.edu.cn (D.C.); gong.qiucheng@nudt.edu.cn (Q.G.); wei.liu.pku@gmail.com (W.L.); xwlong110@sina.com (X.L.)
2   Nanhu Laser Laboratory, National University of Defense Technology, Changsha 410073, China
*   Correspondence: zhqitan@sina.com
†   These authors contributed equally to this work.

**Abstract:** He-Ne lasers play a crucial role in ultra-precision measurement and optical sensing across various fields. For many applications based on He-Ne lasers, a higher output power is required to enhance the accuracy and signal-to-noise ratios of the associated optical measurements. However, conventional methods to increase the output power by reducing the diameter of the He-Ne laser discharge capillary inevitably result in higher diffraction losses and constrain the lasing performance. Here, we propose an approach to enhance laser pumping efficiency and output power through optimizing the ratios of He and Ne gasses. The validity of our proposal has been confirmed by both numerical simulations of He-Ne laser plasma discharge processes and experimental demonstrations, showing that the optimal gas ratio increases with the capillary diameter and total gas pressure.

**Keywords:** He-Ne lasers; pumping efficiency; plasma dynamics; gas ratio

## 1. Introduction

The He-Ne laser has been widely employed for both fundamental scientific explorations and practical optical applications [1–7]. Its ultra-narrow linewidth and high stability make it an indispensable tool in high-precision measurement and have contributed significantly to many optical technological breakthroughs [7–10]. The operation of He-Ne lasers relies on the plasma discharge, during which the helium and neon gas mixture within the laser tube undergoes optical amplification through electron–atom excitation and de-excitation processes [11]. This amplification results in the emission of coherent light at a specific wavelength, typically 632.8 nm, on which many high-precision measurements are based.

Conventionally, the optimization of He-Ne lasers in different devices mainly comes from substantial engineering experience but lacks theoretical calculations. Typically, the gain of a He-Ne laser is enhanced by reducing the diameter of the discharge capillary [12–14]. However, this reduction in diameter leads to a decrease in the volume of the laser mode and consequently also significant diffraction losses. Moreover, the fabrication of narrower capillaries in glass-ceramic materials with ultra-low thermal expansion coefficients is extremely challenging, which restricts the manufacturing of large-scale active interferometers based on He-Ne lasers [15–18].

Some theoretical studies related to He-Ne laser plasma have also been conducted in recent years. Tian Shuangchen et al. employed Comsol Multiphysics to analyze the migration behavior of $Li^+$ ions on the surface of the microcrystalline glass, proposing potential methods to enhance the long-term stability of the He-Ne ring lasers [19]. Zheng Xin et al. simulated and studied the loss changes in a laser resonator under plasma effects

based on the electrons and positive ions distributed on the reflective mirror surfaces inside the cavity [20]. J S Macé et al. established a model of He-Ne plasma discharge and investigated the density distribution of excited state atoms in the anode column of the DC discharge. They found that both radiative and non-radiative diffusion of excited state atoms significantly affect the radial distribution of gain for laser modes [21]. However, the plasma dynamics and atomic state distribution under different He/Ne gas ratios conditions have seldom been studied.

In this study, we establish a plasma dynamics model of He-Ne lasers to numerically study the distribution of He and Ne atoms under DC discharge conditions, and propose a method to increase the gain efficiency of the He-Ne laser operating at 632.8 nm by optimizing the ratios of He and Ne atoms. The plasma dynamics under different capillary diameters, gas pressures and gas ratios are simulated and analyzed using finite element methods. Both those simulations and further experimental demonstrations have verified the validity of our proposal. The effects of the discharge capillary diameter and total gas pressure on the optimal ratios of He and Ne gases are also investigated. Our study would render new opportunities for not only the investigations into the He-Ne lasers themselves but also for enormous applications that rely on He-Ne lasers. It would also shed new light on the more general gas lasers.

## 2. Plasma Simulation of DC Discharge in He-Ne Lasers

Figure 1a shows the structure of a DC-discharge He-Ne laser cavity used in experiments, whose capillary diameter is 6 mm and mode space is 2 GHz. The diaphragm in the laser cavity is used to compress the high-order transverse modes, ensuring the oscillation of the single $TEM_{00}$ mode shown in Figure 1d. Throughout this paper, we focus only on the gain atoms that contribute to the single longitudinal mode operation. No other modes are present in our study, and thus, no effects originating from multimode competitions are taken into consideration. As is shown in Figure 1c, a stable electron distribution can be generated between the cathode and anode by undergoing a breakdown discharge process with a high voltage of 2.4 kV first, and finally maintaining by a low current (~1.6 mA). The stable distribution of electron and He-Ne plasma in this discharge region is the source of laser gain. In this study, a laser resonant cavity consisting of a plane and the spherical mirrors with a curvature radius of 8 m is used. The beam diameter evolutions in the cavity are shown in Figure 1b. We will analyze statistically the distribution of various plasma species and the average reaction rate in this region.

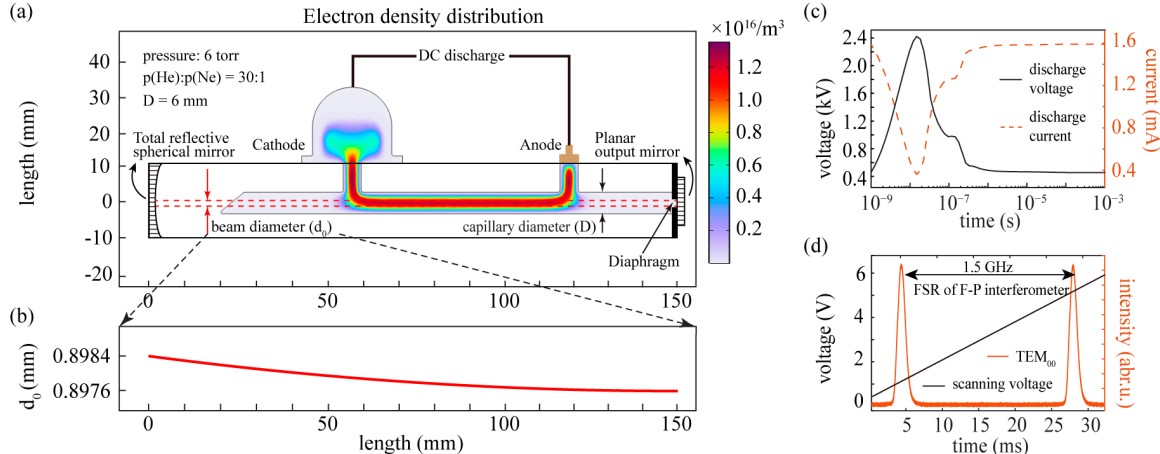

**Figure 1.** Details of a DC-discharge He-Ne laser and its discharge process to a steady state. (**a**) Structure of the He-Ne laser based on DC discharge, and the corresponding distribution of the intrinsic optical field in the resonator (red dashed line); (**b**) beam diameter evolutions of the optical field along the cavity inside the laser; (**c**) the establishment process of steady-state voltage and current in the plasma discharge; (**d**) mode structure of the He-Ne laser obtained by Fabry–Perot Interferometer.

The plasma reactions between He and Ne atoms are highly complex processes. Many processes have small reaction coefficients and do not qualitatively affect the final steady-state results. Therefore, this study focuses on modeling and simulating the plasma dynamics of the four main transition spectral lines (3.39 μm, 1.15 μm, 543.3 nm, and 632.8 nm) in the He-Ne laser, of which the main reaction chemical equations and their corresponding coefficients are shown in Table 1. ($T_g$ and $T_e$ are the temperature of He-Ne mixed gas and electron, respectively.)

**Table 1.** Main reaction equations for the He-Ne laser system [22–30].

| Reactions Types | Reactions Equations | Reactions Coefficients |
|---|---|---|
| Electron-He collisions | $e + \text{He} \rightarrow e + \text{He}$ <br> $e + \text{He} \rightarrow e + \text{He}(2^3\text{S})$ <br> $e + \text{He}(2^3\text{S}) \rightarrow e + \text{He}$ <br> $e + \text{He} \rightarrow e + \text{He}(2^1\text{S})$ <br> $e + \text{He}(2^1\text{S}) \rightarrow e + \text{He}$ <br> $e + \text{He} \rightarrow 2e + \text{He}^+$ <br> $e + \text{He}(2^3\text{S}) \rightarrow 2e + \text{He}^+$ <br> $e + \text{He}(2^1\text{S}) \rightarrow 2e + \text{He}^+$ | Collision section data from NIST |
| | $e + \text{He} + \text{He}^+ \rightarrow \text{He} + \text{He}(2^1\text{S})$ <br> $e + \text{He} + \text{He}^+ \rightarrow \text{He} + \text{He}(2^3\text{S})$ | $(T_g/T_e)^{-2} \times 10^{-20}$ <br> $(\text{mm}^3/\text{s})$ |
| | $\text{He}(2^3\text{S}) + \text{He}(2^3\text{S}) \rightarrow e + \text{He} + \text{He}^+$ <br> $\text{He}(2^3\text{S}) + \text{He}(2^1\text{S}) \rightarrow e + \text{He} + \text{He}^+$ <br> $\text{He}(2^1\text{S}) + \text{He}(2^1\text{S}) \rightarrow e + \text{He} + \text{He}^+$ | $8.7 \times 10^{-7}(T_g/0.025)^{1/2}$ <br> $(\text{mm}^3/\text{s})$ |
| Electron-Ne collisions | $e + \text{Ne} \rightarrow e + \text{Ne}$ <br> $e + \text{Ne} \rightarrow e + \text{Ne}(2p^53\text{S})$ <br> $e + \text{Ne}(2p^53\text{S}) \rightarrow e + \text{Ne}$ <br> $e + \text{Ne}(2p^53\text{S}) \rightarrow 2e + \text{Ne}$ <br> $e + \text{Ne} \rightarrow e + \text{Ne}(2p^55\text{S})$ <br> $e + \text{Ne}(2p^55\text{S}) \rightarrow e + \text{Ne}$ <br> $e + \text{Ne} \rightarrow 2e + \text{Ne}^+$ | Collision section data from NIST |
| He-Ne collisions | $\text{He}(2^1\text{S}) + \text{Ne} \rightarrow \text{Ne}(2p^55\text{S}) + \text{He}$ <br> $\text{He}(2^3\text{S}) + \text{Ne} \rightarrow \text{Ne}(2p^54\text{S}) + \text{He}$ | $4 \times 10^{-8}$ (mm$^3$/s) <br> $4 \times 10^{-9}$ (mm$^3$/s) |
| Ne radiations | $\text{Ne}(2p^55\text{S}) \rightarrow \text{Ne}(2p^54\text{P})$ <br> $\text{Ne}(2p^55\text{S}) \rightarrow \text{Ne}(2p^53\text{P}_{1/2})$ <br> $\text{Ne}(2p^55\text{S}) \rightarrow \text{Ne}(2p^53\text{P}_{3/2})$ <br> $\text{Ne}(2p^54\text{S}) \rightarrow \text{Ne}(2p^53\text{P}_{1/2})$ <br> $\text{Ne}(2p^54\text{P}) \rightarrow \text{Ne}(2p^53\text{S})$ <br> $\text{Ne}(2p^53\text{P}_{1/2}) \rightarrow \text{Ne}(2p^53\text{S})$ <br> $\text{Ne}(2p^53\text{P}_{3/2}) \rightarrow \text{Ne}(2p^53\text{S})$ <br> $\text{Ne}(2p^53\text{S}) \rightarrow \text{Ne}$ | $2.92 \times 10^6$ (s$^{-1}$) <br> $3.39 \times 10^6$ (s$^{-1}$) <br> $2.83 \times 10^5$ (s$^{-1}$) <br> $1.07 \times 10^7$ (s$^{-1}$) <br> $8.70 \times 10^5$ (s$^{-1}$) <br> $1.81 \times 10^7$ (s$^{-1}$) <br> $1.01 \times 10^7$ (s$^{-1}$) <br> $4.40 \times 10^7$ (s$^{-1}$) |
| De-excitations by wall collisions | $\left.\begin{array}{l}\text{Ne}(2p^55\text{S}) \\ \text{Ne}(2p^54\text{S}) \\ \text{Ne}(2p^54\text{P}) \\ \text{Ne}(2p^53\text{P}_{1/2}) \\ \text{Ne}(2p^53\text{P}_{3/2}) \\ \text{Ne}(2p^53\text{S}) \\ \text{Ne}^+\end{array}\right\} \rightarrow \text{Ne}$ <br> $\left.\begin{array}{l}\text{He}(2^1\text{S}) \\ \text{He}(2^3\text{S}) \\ \text{He}^+\end{array}\right\} \rightarrow \text{He}$ | Sticking coefficient is 1 <br> Secondary electron emission coefficient is 0.03 |

As shown in Figure 2a, the He-Ne laser is a typical four-level laser system, where the pumping of Ne atoms originates mostly from the resonant energy transfer from He atoms during the collisions ($S_{03}$). The atoms in the lower energy levels of Ne will rapidly return to the metastable state Ne ($2p^53\text{S}$) through spontaneous or nonradiative transitions ($A_{21}$),

which have a longer lifetime and generally require de-excitation to the ground state through inelastic collisions with the capillary wall ($A_{10}$). Traditionally, compressing the diameter of the He-Ne laser discharge capillary can significantly increase the frequency of inelastic collisions between Ne atoms and the capillary wall, thereby enhancing the laser gain.

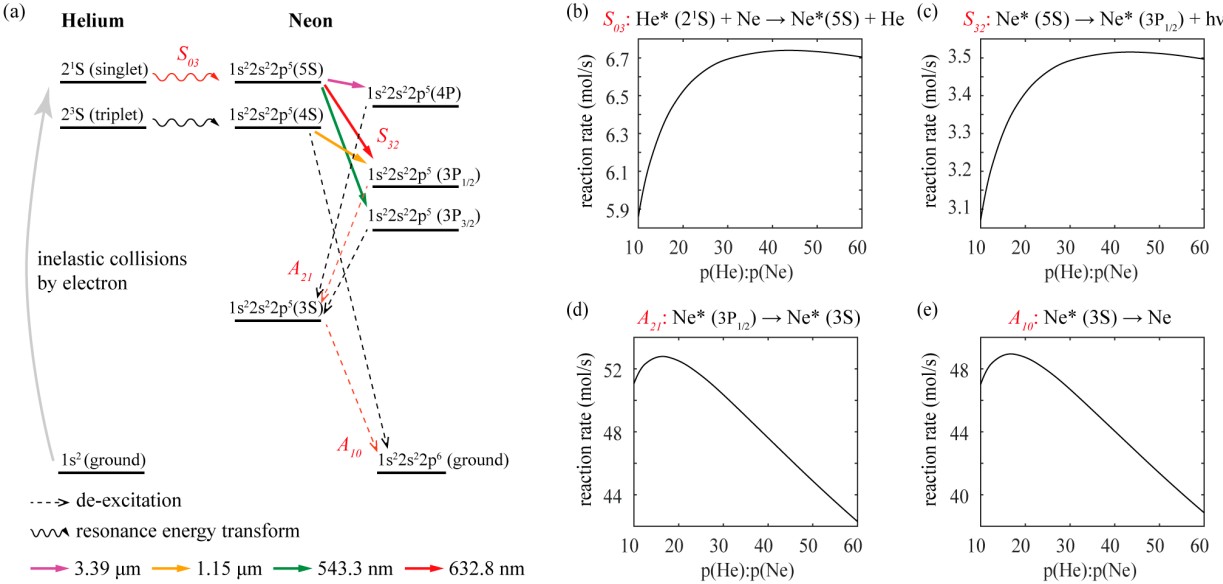

**Figure 2.** The key reactions related to a 632.8 nm laser. (**a**) Energy level diagram of the He-Ne laser, including the major laser spectral lines: 3.39 μm, 1.15 μm, 632.8 nm, and 543.3 nm [31]; (**b–e**) the simulation results of four key reaction rates ($S_{03}$, $S_{32}$, $A_{21}$, $A_{10}$) related to 632.8 nm laser for different He/Ne gas ratios. * represents the atoms in the excited state.

The following four-step plasma reaction equations dominate the generation processes of a laser at the wavelength of rate equations [32]:

$$S_{03} : \ \text{He}^*(2^1\text{S}) + \text{Ne} \rightarrow \text{Ne}^*(2\text{p}^5 5\text{S}) + \text{He} - \Delta E(0.047 \ \text{eV}) \tag{1}$$

$$S_{32} : \ \text{Ne}^*(2\text{p}^5 5\text{S}) \rightarrow \text{Ne}^*(2\text{p}^5 3\text{P}_{1/2}) + h\nu(632.8 \ \text{nm}) \tag{2}$$

$$A_{21} : \ \text{Ne}^*(2\text{p}^5 3\text{P}_{1/2}) \rightarrow \text{Ne}^*(2\text{p}^5 3\text{S}) \tag{3}$$

$$A_{10} : \ \text{Ne}^*(2\text{p}^5 3\text{S}) \rightarrow \text{Ne} \tag{4}$$

In the above equations, * represents the atoms in the excited state. Equation (1) describes the inelastic collisions between He and Ne atoms, which is also named as resonance energy transfer; Equation (2) describes the spontaneous radiation process of Ne atoms, which produces photons with a wavelength of 632.8 nm, serving as the basis of laser generation; Equations (3) and (4) describe the de-excitation process of Ne, where the excited state Ne*($2\text{p}^5 3\text{P}_{1/2}$) has a short lifetime and can quickly return to the metastable state Ne*($2\text{p}^5 3\text{S}$) mainly by spontaneous radiation, while the metastable state Ne*($2\text{p}^5 3\text{S}$) return to the ground state mainly through inelastic collisions with the capillary wall. The rates of these reactions and the number of particles in various excited states of Ne will affect the final laser output power.

It can be observed from Figure 2c–e that the average reaction rates described by Equations (1)–(4) increase rapidly with the increase in the He/Ne gas ratio initially, while the de-excitation processes of Ne*($A_{21}$ and $A_{10}$) decrease dramatically with higher He/Ne gas ratio. In a laser reaction circle, higher excitation rates ($S_{03}$) can pump more atoms to the higher energy levels, while lower excitation rates ($A_{21}$ and $A_{10}$) will cause atoms to remain at low energy levels, thus destroying the population inversion of a laser. These simulation

results reflect the tradeoff between them and show the existence of an optimal gas ratio for Ne atoms emitting the 632.8 nm laser.

The distributions of different excited atoms related to 632.8 nm laser under various He/Ne gas ratios are also studied. The insets in Figure 3a,b show the spatial distributions of excited atoms He*($2^1$S) and Ne*($2p^55$S, $2p^53P_{1/2}$, $2p^53$S) with a He/Ne gas ratio of 30:1 under steady-discharge conditions, where the discharge capillary width is 6 mm and the total gas pressure is 6 torr. It is clear that the distribution of the excited atoms is highly consistent with the electron distribution shown in Figure 1a. According to Equations (1)–(4), a sufficient atom population of He*($2^1$S) is required to excite the 632.8 nm laser of Ne. As shown in Figure 3a, although the proportion of He*($2^1$S) among all He excited states is relatively low (<2%) under steady-state conditions, it increases linearly with the increasing He/Ne gas ratio, indicating that more He atoms are involved in the production process of the 632.8 nm emission. On the other hand, as shown in Figure 3b, although the total number of Ne atoms decreases, the total number of excited Ne atoms responsible for the 632.8 nm laser emission exhibits an initial increase followed by a decrease with increasing He/Ne gas ratios. This indicates the presence of an optimal He/Ne gas ratio that maximizes the number of participating Ne atoms. Finally, combining all the simulation results of key reaction coefficients and atom distributions, the relationship between the inverted population and the He/Ne gas ratio can be obtained by numerically solving the following rate equations [29]:

$$
\begin{cases}
\frac{dN_3}{dt} = N_0 S_{03} - N_3 S_{32} - (N_3 - N_2) N_\nu S_\nu \\
\frac{dN_1}{dt} = N_2 A_{21} - N_1 A_{10} \\
\frac{dN_\nu}{dt} = (N_3 - N_2) N_\nu S_\nu - \frac{N_\nu}{\tau} \\
N_0 S_{03} = N_1 A_{10} \\
N = N_1 + N_2 + N_3
\end{cases} \tag{5}
$$

where $N_0$~$N_3$ are the number of Ne (ground state), Ne*($2p^53$S), Ne*($2p^53P_{1/2}$), and Ne*($2p^55$S), respectively; $N$ is the total number of excited Ne atoms used in producing 632.8 nm laser; $N_\nu$ is the photon number and $S_\nu$ is the stimulated radiation rate; $S_{03}$, $S_{32}$, $A_{21}$, and $A_{10}$ are reaction rates described in Equations (1)–(4); and $\tau$ is the lifetime of the photon confined in the laser resonator. To simplify the rate equations, we make the following approximation: (1) the influence of other plasma reactions on the number of Ne*($2p^55$S, $2p^53P_{1/2}$, $2p^53$S) is neglected; (2) the de-excited atom number from Ne*($2p^53$S) and excited atom number from Ne ground state are in the dynamic equilibrium state. The inverted population can be obtained by ($N_3 - N_2$) from the equilibrium solution of Equation (5).

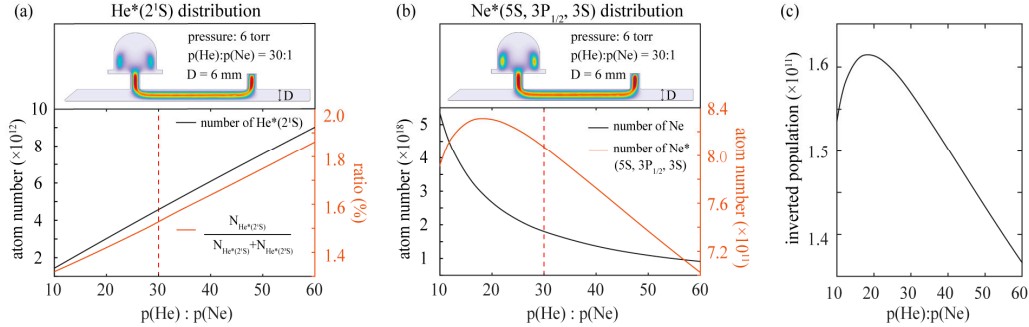

**Figure 3.** The number of excited atoms and inverted population simulated for different He/Ne gas ratios. (**a**,**b**) Variations in the number of different excited atoms with He/Ne gas ratios under steady-discharge conditions. The insets in (**a**,**b**) illustrate the simulated distribution of excited atoms in a laser with p(He):p(Ne) = 30:1, which is also indicated by red dashed lines. (**c**) Variations in the population inversion with He/Ne gas ratios of the laser with a total pressure of 6 torr and capillary diameter of 6 mm. * represents the atoms in the excited state.

As shown in Figure 3c, there exists a peak value of the inverted population with different He/Ne gas ratios. This shows that under the conditions of a discharge capillary diameter of 6 mm and a gas pressure of 6 torr, a He/Ne gas ratio of 18.3:1 could achieve the highest laser gain efficiency and maximum output power for the 632.8 nm emission.

### 3. Optimal He/Ne Gas Ratio

It is worth noting that different optimal He/Ne gas ratios can be obtained under different experimental conditions according to previous works [33–35], which may be influenced by both the diameter of the discharge capillary and the total pressure. As shown in Figure 4, simulation results indicate that as the pressure (horizontal axis) and capillary diameter (vertical axis) increase, the optimal He/Ne gas ratio corresponding to the maximal inverted population also increases gradually. This suggests that discharge tubes of larger cross sections designed to reduce diffraction losses and increase laser mode volume typically require higher He/Ne gas ratios (>20:1) to obtain the optimum gain.

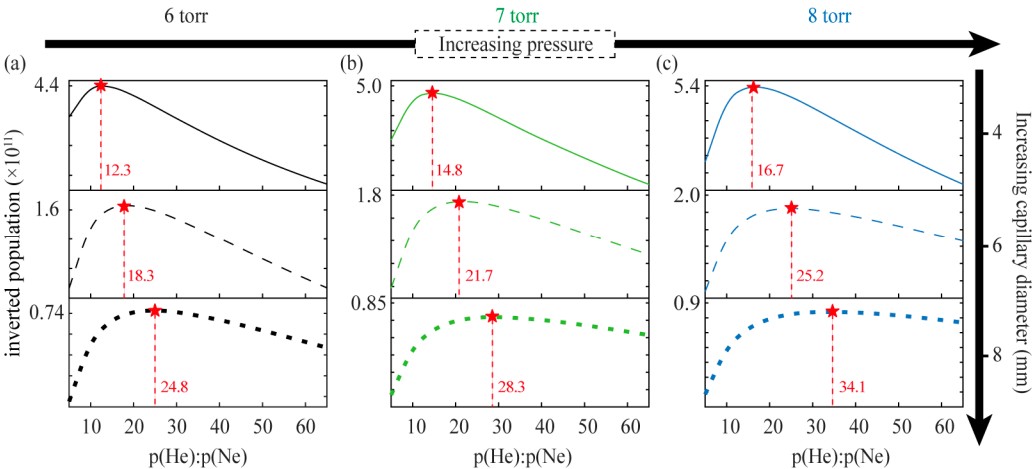

**Figure 4.** The dependence of the optimal He/Ne gas ratio on the diameter of the discharge capillary under different pressures. (**a–c**) Summarize simulations under pressures of 6 torr, 7 torr, and 8 torr, respectively. The red stars indicate the maximum inverted population and the optimal He/Ne gas ratio under different conditions.

To validate the simulation model presented in this paper, experiments with different He/Ne gas ratios are conducted. In our experiment setup, the diameter of the discharge capillary is 6 mm and the discharge length is 60 mm as is shown in Figure 1a, which is the same as the parameters adopted in the previous simulations. As shown in Figure 5a,b, under low-gain conditions, there is a linear relationship between laser output power and pump power, and the slopes of these lines differ significantly under different gas ratios. We define these slope values as the gain efficiency [as demonstrated in Figure 5c], which exhibits a significant increase between He/Ne gas ratios of 10:1 and 30:1. Figure 5d further demonstrates that under the same pump conditions, higher He/Ne gas ratios are required to achieve maximum output power as the pressure increases, which is consistent with the simulation results shown in Figure 4.

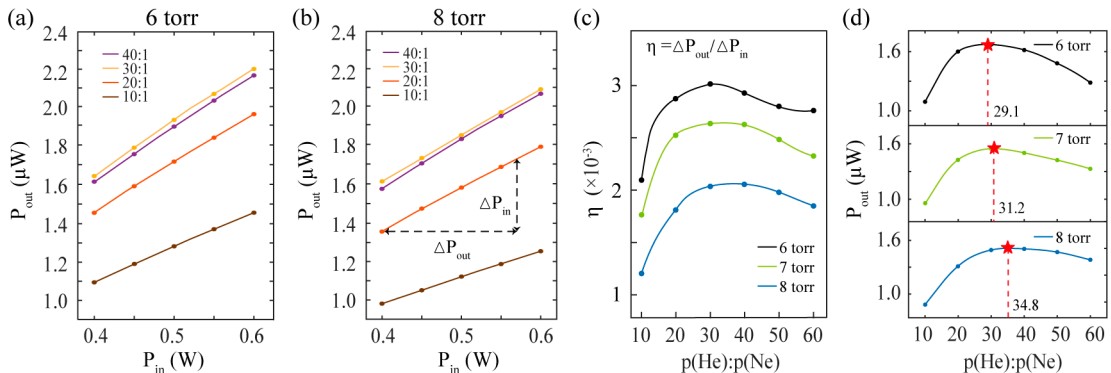

**Figure 5.** Experiment results of gain efficiency and output power of a He-Ne laser under different gas ratios and pressure conditions. (**a**,**b**) The effects of different He/Ne gas ratios on the correlation between laser output power and electrical pump power for gas pressure of 6 torr and 8 torr, respectively. (**c**) The variation in electrical pump efficiency with He/Ne gas ratio under different pressures. (**d**) The optimal He/Ne gas ratio under the same pump conditions but different pressures. The red stars indicate the maximum output power and optimal He/Ne gas ratio under different pressure conditions.

## 4. Discussion and Conclusions

Through the simulation of the DC-discharged He-Ne plasma system, we have found that different He/Ne gas ratios significantly affect the total number of excited Ne atoms, which is a key factor in determining the optimal He/Ne gas ratio under the same gas pressure and capillary diameter conditions. Furthermore, the de-excitation process of Ne atoms from the laser lower level is highly correlated with the He/Ne gas ratio. With an increase in the proportion of He atoms, the de-excitation rate of Ne atoms decreases significantly, resulting in a decrease in the inverted populations.

Under the conditions of a capillary diameter of 6 mm and a total pressure of 6 torr, the experimental results of the optimal He/Ne gas ratio are slightly higher than the simulation results. The main reasons for this difference are as follows: (1) The simulation model employs a quasi-three-dimensional structure (as introduced in Appendix A), resulting in a reduction in the electron emission area for semi-spherical discharge cathode. (2) In this study, only the reaction processes related to the main spectral lines of 3.39 μm, 1.15 μm, 632.8 nm, and 543.3 nm were selected, while other numerous non-radiative transition processes were neglected. Furthermore, in actual laser processes, Ne atoms release energy through photon emission, which cannot be considered in the plasma simulation. These approximations will affect the atomic state and energy distributions of Ne atoms under steady-state conditions, leading to the difference in the value of the optimal gas ratio in simulations and experiments. In the future, we will establish a more complete 3D simulation structure under sufficient computing power to achieve more accurate results.

In conclusion, we have established a simulation model for the direct current discharge process of He-Ne laser plasma based on finite element methods. The plasma dynamics under different pressures, gas ratios, and discharge capillary diameter conditions have been systematically analyzed. It is revealed that in the discharge process to produce the 632.8 nm laser, the inverted population can be enhanced by tuning the He/Ne gas ratio. Further computational simulations and experimental results have confirmed that increasing the discharge capillary diameter and total pressure inside the cavity requires higher He/Ne gas ratios to achieve optimum pump efficiency and maximum output power. This study has significantly expanded the horizons for the design and production of more efficient He-Ne lasers, as well as related laser interferometers and precision measurement devices. It enables the enhancement of the He-Ne laser output power without modifying the laser resonator or discharge structure, thereby reducing quantum noise in ultra-precise sensors such as ring laser gyroscopes [7,36,37]. Our study could potentially stimulate many further explorations into gas laser dynamics and their practical applications.

**Author Contributions:** Conceptualization, Y.M. and J.H.; methodology, Y.M., H.J. and J.H.; software, Y.M. and D.C.; validation, D.C., Q.G. and W.L.; formal analysis, Y.M. and J.H.; investigation, Y.M. and S.G.; resources, X.L.; data curation, X.L. and Z.T.; writing—original draft preparation, Y.M.; writing—review and editing, W.L.; visualization, Y.M.; supervision, W.L.; project administration, Z.T.; funding acquisition, Z.T. All authors have read and agreed to the published version of the manuscript.

**Funding:** Science Foundation for Indigenous Innovation of National University of Defense Technology (22-ZZCX-063); Natural Science Foundation of Hunan Province of China (2023JJ30639); Natural Science Foundation of China (62375285).

**Data Availability Statement:** The authors declare the data that support the findings of this study are available upon reasonable request.

**Conflicts of Interest:** The authors declare no conflicts of interest.

### Appendix A. Methods of He-Ne Plasma DC-Discharged Modelling

In this paper, the plasma module in COMSOL Multiphysics® 6.1 (a commercial simulation software based on the finite element method) is used to simulate the glow discharge process of the He-Ne laser. The modeling process and parameter settings are as follows:

**Table A1.** Parameter settings for the He-Ne plasma dynamics simulations.

| Parameters | Value | Description |
|---|---|---|
| $\gamma_e$ | 0.03 | secondary electron emission coefficient |
| $W_f$ | 5 (eV) | wall work function |
| $M_{He}$ | 0.004 (kg/mol) | molar mass of He |
| $\sigma_{He}$ | 2.58 (Å) | potential characteristic length of He |
| $M_{Ne}$ | 0.02 (kg/mol) | molar mass of Ne |
| $\sigma_{Ne}$ | 2.82 (Å) | potential characteristic length of Ne |
| $n_e$ | $10^{14}$ (1/m$^3$) | initial electron density |
| $T_g$ | 300 (K) | gas temperature |

To begin with, a two-dimensional model of the cross-section of the He-Ne laser discharge tube is constructed as shown in Figure 1a, and the depth of the discharge region is set to the diameter of the capillary to achieve quasi-three-dimensional computational results. Then, the discharge region is divided into mesh grids, where we particularly pay attention to the treatment of the boundary region. There are significant de-excitation reactions at the walls of the discharge tube, where plasma species in excited or ionized states return to the ground state through collisions with the boundaries.

The result of the mesh grid division is shown in Figure A1, with an inserted detail illustrating the grid division at a corner boundary. To avoid singular points in the calculations, all corner regions are subjected to rounding treatment, and an eight-layer quadrilateral grid is used to transition from the boundary to the interior of the discharge region, ensuring the validity of the calculations for wall collision reactions.

After completing the above modelling, the reaction processes listed in Table 1 in the main text are set into the plasma discharge region, the shaded area in Figure A1. Finally, a temporal analysis solver is used to calculate the discharge evolution process of the plasma within the range of $10^{-9}$ to $10^{-3}$ s, typically converging to a steady-state solution within the range of $10^{-6}$ to $10^{-5}$ s. Finally, the distributions of all species and steady-state reaction rates can be obtained.

Throughout the entire computational process, the value of the secondary electron emission coefficient and the wall work function are set as constants, ignoring their relationships with gas pressure and temperature. These assumptions are reasonable within the range of room temperature and small pressure variations (6–8 torr).

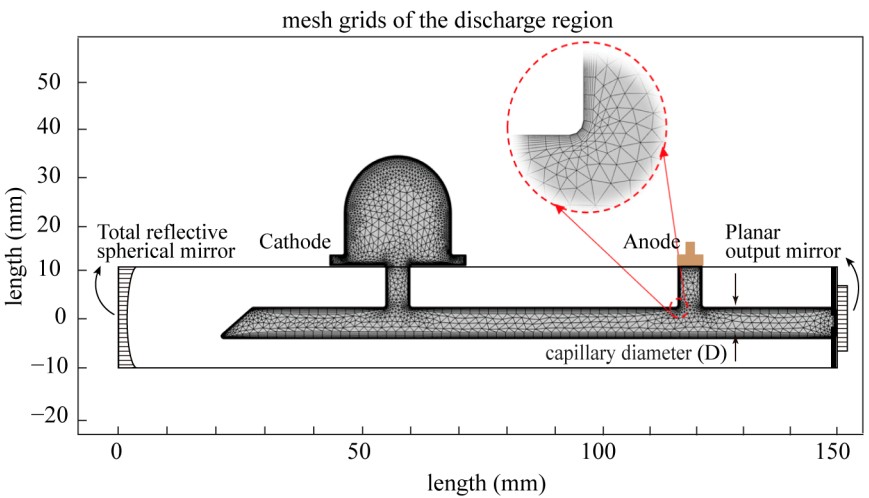

**Figure A1.** Mesh grids in the plasma discharge region.

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
