# Peer review of "Analysis of Plasma Dynamics in He-Ne Lasers with Different Gas Ratios"

_photonics, doi:10.3390/photonics11030276_

Round 1
Reviewer 1 Report
Comments and Suggestions for Authors
In this paper, some interesting plasma dynamic information was obtained through experiments and simulations.
Comments:
(1) How the authors presented their data can be improved further. For example, Table 1 and Table A1 should be integrated into one Table with some sidelines or, at least, placed back-to-back as Table 1 and 2. I assume the numbers and letters after He or Ne are the term symbols. If it is, the presentation is a disaster, worse than not showing them. Please ask some physicists or chemists to assist the authors in writing the correct term symbols and orbital notations. The same problems are also scattered everywhere in the paper. You don’t indicate a reaction or transition from a term symbol to an orbital, or from an orbital to unknown destinations.
(2) The simulation is a black box. It is readable but needs to be decoded first. Not all readers are familiar with finite element methods. The authors only vaguely mentioned the simulation tool, but it’s so mysterious that they don’t discuss what key parameters were used in the simulation. Which parameter is the most sensitive one? How close is the simulated result as compared to the experimental results? No deviation at all? Is there any space for improvement in the current model?
(3) Since the authors have noticed that the tube diameter is an important factor for the optimal performance of He-Ne lasers, another diameter-sensitive factor in plasma should also be considered, ambipolar diffusion and wall reactions. For He and Ne, the most likely major type of the wall reaction will be the de-excitation reaction, which is more sensitive as the pressure increases.
This paper needs a large-scale overhaul. It is not a rejection, but return the manuscript to the authors and ask them to resubmit.
Comments on the Quality of English Language
writing is fair
Reviewer 2 Report
Comments and Suggestions for Authors
In the manuscript “Analysis of plasma dynamics in He-Ne lasers with different gas ratios” submitted by Yuanhao Mao, Jing Hu, Hongteng Ji, Shiyu Guan, Dingbo Chen, Qiucheng Gong, Wei Liu, Xingwu Long, Zhongqi Tan, the authors demonstrate a plasma dynamics model in He-Ne lasers which can predict the gain efficiency under different capillary diameters, gas pressures and gas ratios. The experimental results are consistent with simulation predictions in some respects.
The authors first introduce the He-Ne laser system. With a plasma simulation, the authors achieve the reaction rate of key reactions related to the 632.8 nm laser. The numerical solutions of the rate equations are introduced in detail. Combining those simulation techniques, the authors explore the optimal He/Ne gas ratio for high gain efficiency with different pressures and capillary diameters. Finally, the experiment is introduced. It shows a trend of gain efficiency that is consistent with the simulation.
Overall, the manuscript is well written with good data presentation and reasonable arguments. It is nice to see the experimental validation. Some more introductions to the simulation methods could make the work more complete. Some points need more careful explanations. This manuscript can be published with the consideration of the following remarks:
(1) In the introduction part, the authors mention the background of He-Ne lasers. I would suggest the authors include the investigation of previous works on the He-Ne laser simulation. Compared with previous works, the readers will have a better understanding in the advance of this paper.
(2) In Figure 2 (b)-(e), the authors show the simulation results of reaction rates of all four key reactions for 632.8 nm lasering. How are these figures simulated and generated? What are the simulation parameters? Which software do you use? Did you make any assumptions? Please include a brief introduction about the simulation method for this setup and the corresponding parameters used. It is easy to be confused with the second numerical simulation.
(3) On page 6, the authors present the experiment methods and results. The experiment shows that with 6 mm capillary and 6 torr pressure, the optimal HeNe ratio is about 29.1. However, for the simulation, it is predicted to be 18.3. Could the authors discuss the possible causes of this discrepancy? Please include this discussion in the manuscript.
(4) In the conclusion part, the authors summarize the content of the manuscript. Please also consider expanding the discussion on the potential uncertainties, limitations, and future directions of this simulation model. This will help the readers better understand the scope of this manuscript.
Reviewer 3 Report
Comments and Suggestions for Authors
I would like to congratulate the authors for their work. I only have a suggestion for expanding the writing to provide further insights into the significance of these research results. Could you make specific recommendations for further study, investigation, and applications?
Reviewer 4 Report
Comments and Suggestions for Authors
This is interesting and unexpected look at the performance of the most established gas laser system He Ne. it is shown how pressure and diameter of the gas discharge tube defines the inversion conditions. Hence, laser performance.
It is a good study and clearly explained. Improvements could be made to add to the explanation related to the stability of the laser performance. Stability of inversion might have slightly different conditions as the inversion alone.
How do inversion conditions affect coherence length?
Round 2
Reviewer 1 Report
Comments and Suggestions for Authors
Comsol Multiphysics is a software, not an instrument. Not many readers know what it is. Please check with the editor of this journal about the correct format for describing software or instruments. Add some info, such as publishing company, version, etc.
No other issues for now.
Author Response
We thank reviewer 1 for the valuable suggestions. We have revised Appendix A for a clearer description of the software we used in this paper:
Appendix A: Methods of He-Ne plasma DC-discharged modelling
In this paper, the plasma module in COMSOL Multiphysics® 6.1 (a commercial simulation software based on the finite element method) is used to simulate the glow discharge process of the He-Ne laser.
